# Terahertz Time-Domain Spectroscopy of Glioma Patient Blood Plasma: Diagnosis and Treatment

Olga Cherkasova [1,2,*], Denis Vrazhnov [3,4], Anastasia Knyazkova [3,4], Maria Konnikova [2,3,5], Evgeny Stupak [6], Vadim Glotov [6], Vyacheslav Stupak [6], Nazar Nikolaev [1], Andrey Paulish [7,8], Yan Peng [9], Yury Kistenev [3,4] and Alexander Shkurinov [3,5]

1   Institute of Automation and Electrometry, Siberian Branch of the Russian Academy of Sciences, 630090 Novosibirsk, Russia; nazar@iae.nsk.su
2   Institute on Laser and Information Technologies, Branch of the Federal Scientific Research Centre "Crystallography and Photonics" of RAS, 140700 Shatura, Russia; konnikova.mr20@physics.msu.ru
3   Laboratory of Laser Molecular Imaging and Machine Learning, Tomsk State University, 634050 Tomsk, Russia; denis.vrazhnov@gmail.com (D.V.); a_knyazkova@bk.ru (A.K.); yuk@iao.ru (Y.K.); ashkurinov@physics.msu.ru (A.S.)
4   V.E. Zuev Institute of Atmospheric Optics SB RAS, Academician Zuev Square, 1, 634055 Tomsk, Russia
5   Faculty of Physics, Lomonosov Moscow State University, 119991 Moscow, Russia
6   Novosibirsk Research Institute of Traumatology and Orthopedics n.a. Ya.L. Tsivyan, 630091 Novosibirsk, Russia; stupakphoto@mail.ru (E.S.); glotovvadim1998@gmail.com (V.G.); alexdok2000@gmail.com (V.S.)
7   Novosibirsk Division of Rzhanov Institute of Semiconductor Physics Siberian Branch of the Russian Academy of Sciences "Technological Design Institute of Applied Microelectronics", 630090 Novosibirsk, Russia; paulish63@ngs.ru
8   Faculty of Radio Engineering and Electronics, Novosibirsk State Technical University, Karl Marks Avenue, 20, 630073 Novosibirsk, Russia
9   Terahertz Biomedical Laboratory, University of Shanghai for Science and Technology, 516 Jungong Road, Yangpu District, Shanghai 200093, China; py@usst.edu.cn
*   Correspondence: cherkasova@laser.nsc.ru or o.p.cherkasova@gmail.com

**Abstract:** Gliomas, one of the most severe malignant tumors of the central nervous system, have a high mortality rate and an increased risk of recurrence. Therefore, early glioma diagnosis and the control of treatment have great significance. The blood plasma samples of glioma patients, patients with skull craniectomy defects, and healthy donors were studied using terahertz time-domain spectroscopy (THz-TDS). An analysis of experimental THz data was performed by machine learning (ML). The ML pipeline included (i) THz spectra smoothing using the Savitzky–Golay filter, (ii) dimension reduction with principal component analysis and t-distribution stochastic neighborhood embedding methods; (iii) data separability analyzed using Support Vector Machine (SVM), Random Forest (RF), and Extreme Gradient Boosting (XGBoost). The ML models' performance was evaluated by a k-fold cross validation technique using ROC-AUC, sensitivity, and specificity metrics. It was shown that tree-based ensemble methods work more accurately than SVM. RF and XGBoost provided a better differentiation of the group of patients with glioma from healthy donors and patients with skull craniectomy defects. THz-TDS combined with ML was shown to make it possible to separate the blood plasma of patients before and after tumor removal surgery (AUC = 0.92). Thus, the applicability of THz-TDS and ML for the diagnosis of glioma and treatment monitoring has been shown.

**Keywords:** terahertz time-domain spectroscopy; machine learning; glioma; human blood plasma; Support Vector Machine; Random Forest; Extreme Gradient Boosting

## 1. Introduction

The World Health Organization and the health departments of governments of all countries have listed tackling cancer as a top priority [1]. The main problem is late diagnosis, when the prognosis for a cure is unfavorable. Gliomas are one of the most grievous

malignant central nervous system tumors, and have a high mortality rate with a low survival rate, with severe disability and increased risk of recurrence [2,3]. Glioblastoma multiforme (GBM) is one of the most aggressive, fast-moving, and deadliest types of gliomas [4–6]. According to the American Reggae Center for Tumors of the Nervous System statistics from the 2015–2019 period, the glioblastoma incidence was 14.2% among all tumors of the central nervous system and 50.1% among all malignant brain tumors [3].

Over the past few years, magnetic resonance imaging (MRI) has been the primary diagnostic tool for gliomas [7–9]. Fluid attenuated inversion recovery (FLAIR) imaging is a variant of MRI which provides high-quality tumor visualization. A combination of FLAIR MRI with the specific topological and textural features, and an automatic interpretable machine learning algorithm, provided a 97% accuracy of glioma detection and up to 95% accuracy on the glioma segmentation [10]. Computed tomography [11] and positron emission tomography [9,12] are also used for glioma non-invasive diagnosis. These methods usually visualize gliomas of several centimeters of size [13] that do not correspond to the early tumor stage. The examination of the intraoperatively removed tumor or fine needle aspiration biopsy, using traditional histological, cytological, and histochemical methods, is used to establish or confirm a diagnosis [14]. However, histopathological features may be heterogeneous within a tumor, resulting in potential sampling errors and underestimating malignant biological behavior [6].

A GBM tumor contains small cells, characterized by polymorphism, anaplasia, and indistinct cellular borders, which have specific molecular patterns. Various molecular genetic features of gliomas have been discovered. Isocitrate dehydrogenase1/2 (IDH1/2) mutations, 1p and 19q chromosomal deletions, and the methylation of $O^6$-methylguanine-DNA methyltransferase were among the first discovered glioma biomarkers [15]. The accumulation of information about the genetic and molecular pathways of glioma development has led to a revision of the classification of CNS tumors based on molecular alterations [2,6]. The glioma development molecular pathways include [16–18]:

1. The dysregulation of the signaling pathway implemented through growth factors based on the amplification and mutation of tyrosine kinase receptors (TKRs). TKRs present a heterogeneous group of membrane proteins interacting with epithelial growth factor, vascular endothelial growth factor, and platelet growth factor. TRKs also interact with cytokines, hormones, and other signaling systems;
2. The activation of phosphotidylinositol-3-OH kinase (PI3K)/AKT/mTOR the intracellular signaling pathway ensures cell survival under certain conditions;
3. The inactivation of P53 protein and retinoblastoma suppression pathways;
4. Ras/Raf/MEK/ERK signaling pathway;
5. The regulation of cellular responses by protein kinase C (PKC), including protein secretion, gene expression, and cell proliferation;
6. Cell progression through pRB pathway suppression;
7. $O^6$-methylguanine-DNA methyltransferase (MGMT);
8. TGF-β signaling [19].

The detection of molecular markers in tissues using genomic, proteomic, and metabolomic methods allows for a more accurate diagnosis of glioma in a small volume biopsy sample [2,6,20]. However, these methods are labor intensive and expensive. The tissue biopsy procedure is traumatic and cannot be used to control tumor treatment.

As an alternative to tissue biopsy, the concept of a biofluid biopsy was introduced [21,22]. The molecular composition of body fluids changes significantly with the tumor appearance and development [23–25], which can be a good diagnostic criterion for early cancer diagnosis. Body fluid glioma molecular markers include circulating tumor DNA and RNA, proteins, and metabolites [22,25]. Most of these markers have specific dielectric characteristics in the terahertz (THz) range, which can potentially be used for the early diagnosis of gliomas [23,26]. THz technologies have already shown their applicability in noninvasive diagnostics of various pathologies [27–29]. Many molecular markers, including water [30], have spectral imprints in the THz range, providing a differentiation of healthy tissues from

neoplasms [31–35]. The combination of terahertz time-domain spectroscopy (THz-TDS) and machine learning (ML) has made it possible to perform differential diagnostics of thyroid nodule malignancy by analyzing patients' blood plasma [36]. A spectral analysis of exhaled air and blood plasma by THz spectroscopy showed the promise of this approach for diabetes diagnosis [37,38]. The analysis of saliva from patients with oral lichen planus (OLP) using THz-TDS also showed a good separation between the control group and those with OLP [39]. The difference in the THz spectra of rats' blood serum in the dynamics of experimental liver cancer and its correlation with changing the blood biochemical composition have been established [40]. The possibility of detecting glioma stages through mouse blood serum analysis using THz-TDS has been demonstrated [41].

THz absorption spectra do not have sharp peaks [30,36,38,40] and are difficult to interpret using mathematical statistics methods. ML is a powerful technique for THz data characterization, including the differentiation of datasets with slightly differing THz spectra. The typical machine learning pipeline (MLP) includes steps aimed at noise reduction, outlier removal, informative features selection/extraction, prediction data model creation, etc. [41,42]. Outlier removal is usually performed with the isolation forest method [43]. The drawback of the latter is associated with threshold value choice, which is not trivial. Another method for outlier detection is a visual analysis by t-distribution stochastic neighborhood embedding (t-SNE) [44]. The advantage of the latter is the possibility of checking similarity in the original data before their dimension reduction [45], contrary to principal component analysis (PCA).

Supervised ML methods allow the selection of the most informative features in high dimensional spectral data [41,46]. The choice of supervised ML methods is conditioned by the requirements of model interpretability. The top-scored performance algorithms are Support Vector Machine (SVM), Random Forest (RF), Extreme Gradient Boosting (XGBoost) and Artificial Neural Networks (ANNs) [47]. ANN lacks the explainability of constructed data models. To achieve it, a distillation approach [48] or external methods such as SHapley Additive exPlanations (SHAP) are used [49]. In turn, SHAP is limited by a huge computational time for high-dimensional data [50]. Sequential feature selection methods have poor performance on a high dimensional and inter-correlated data [51]. On the other hand, SVM and tree-based algorithms provide reliable embedded feature selection [52–54].

In this work, we performed a comparative study of the blood plasma of glioma patients and patients with skull craniectomy defects (SCD), as well as healthy donors, using THz-TDS and ML. We investigated the applicability of the ML model initially designed for the separation of the blood plasma THz spectra of experimental U87 glioblastoma and healthy mouse groups [41] to human patients with glioma, and compared the performances of RF, XGBoost, and SVM for THz spectra analysis. The blood plasma samples of glioma patients before and after tumor removal surgery were also examined.

## 2. Materials and Methods

### 2.1. Samples

The study was conducted according to the guidelines of the Declaration of Helsinki; each patient signed informed consent, and the clinical data were depersonalized. The Ethics Committee of the Novosibirsk Research Institute of Traumatology and Orthopedics n.a. Ya.L. Tsivyan approved the study protocol (Permission #004/22-1, 17 January 2022). Glioma patients (WHO Grade 2–4, $n = 14$), patients with SCD ($n = 13$), and healthy donors ($n = 6$) were provided by the Novosibirsk Research Institute of Traumatology and Orthopedics n.a. Ya.L. Tsivyan. Peripheral blood samples were collected in vacutainers with EDTA (Sarstedt AG & Co. KG, Nümbrecht, Germany). The plasma was separated by centrifugation at $2800 \times g$ for 15 min at +4 °C. Blood plasma samples were frozen and stored at −80 °C.

### 2.2. THz Spectroscopy

The blood plasma of patients with glioma, healthy donors, and patients with SCD was studied using the THz time-domain spectrometer T-Spec 800 Teravil (EKSPLA, Vilnius, Lithuania) [55]. The spectrometer has a dynamic range > 90 dB at 0.4 THz and with two delay lines, fast and slow, provides real-time data acquisition with 10 spectra/s speed and spectral resolution up to 1 GHz. Plasma samples were investigated in transmission mode (Figure 1) using the plastic cuvettes [41,56]. These cuvettes had sufficient transparency in the THz spectral range [41], and 0.5 mm thick inner gap cuvettes printed on a Designer X PRO 3D printer (PICASO 3D, Russia) made of Watson plastic (BestFilament, Tomsk, Russia) were used [41].

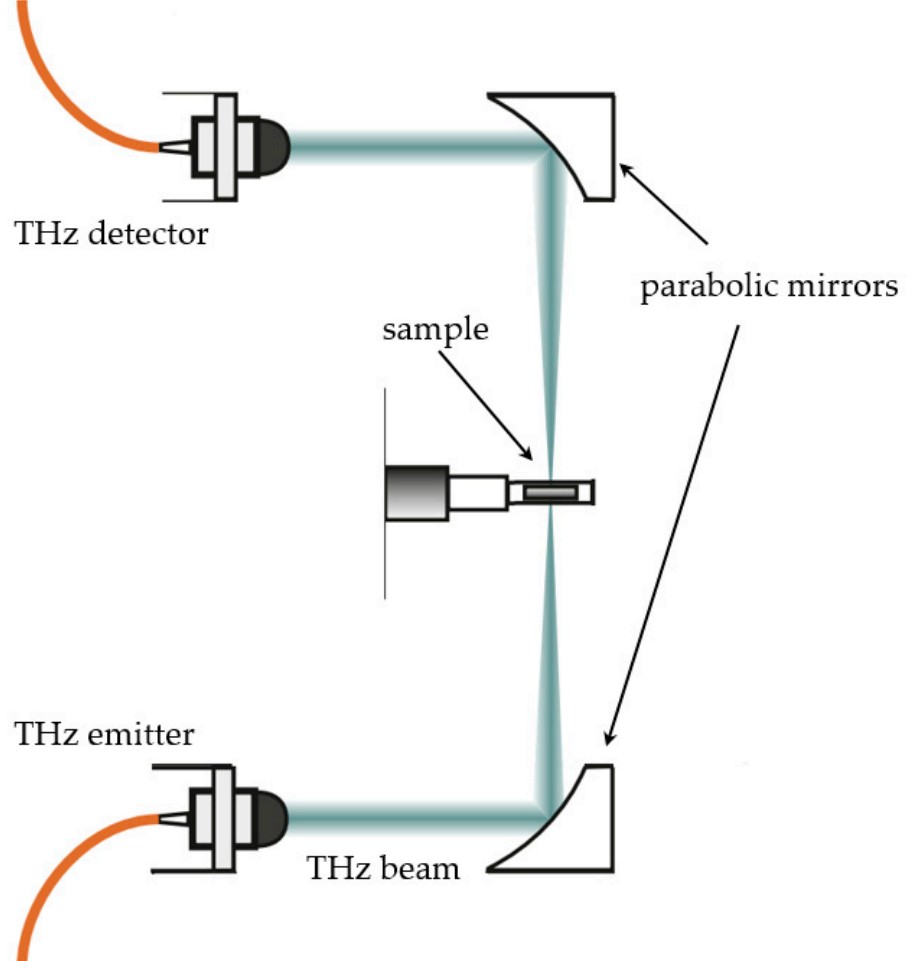

**Figure 1.** Scheme of the cuvette position in the THz time-domain spectrometer. The gray color shows the path of the THz radiation beam through the cuvette center.

An empty cuvette was placed in the focus area of the THz radiation between the parabolic mirrors of the THz-TDS spectrometer, as shown in Figure 1, and a reference signal was recorded. Without changing the position of the cuvette, samples of pre-thawed blood plasma with a volume of 50 μL were placed in the cuvette using an automatic dispenser.

Spectral scanning of the sample was carried out along six spatial points of the cuvette with a step of 0.1 mm vertically and horizontally in the spectral range of 0.2–1.6 THz. Time averaging (over 256 spectra) was performed for each spatial point of the 2D scan. The conversion of the signal from the time domain to the frequency domain was carried out using the Terravil TRS-16 software (TeraVil Ltd., Vilnius, Lithuania). All measurements were carried out at a temperature of 21 ± 1 °C.

### 2.3. Machine Learning Methods

The analysis of the experimental THz data was performed according to the ML pipeline, as presented in Figure 2. In the first stage, data were split into two groups: patients with glioma versus healthy donors (Case 1), and patients with glioma versus patients with SCD (Case 2). A computational processing pipeline was implemented on Python (3.9; here and below, version of the software placed in round brackets) with scikit-learn (1.2.1), scipy (1.9.1), and catboost (1.0.6) libraries. It included THz spectra smoothing using a Savitzky–Golay filter, dimensionality reduction and visualization using PCA and t-SNE methods, and data separability tests using SVM, RF, and XGBoost. The choice of the latter was restricted by the requirement of interpretability of the results. PCA allows the reduction of the dimensionality of the original data by projecting it onto the new coordinate axes (principal components, PCs) by maximizing explained variance. The major difference between PCA and t-SNE methods is that PCA can be trained and then reapplied on the new, unseen data, whereas t-SNE needs to relearn. The advantage of t-SNE is the consideration of distances in the original data. This means that if spectra are close in some metrics in the input data, then after projecting the data to a lower dimension space, points will be close too. It should be mentioned that t-SNE Python implementation (sklearn.manifold.TSNE) has crucial parameters: perplexity and metrics. RF and XGBoost methods are both tree-based ensemble classifiers [57]. The difference between them lies in the training process. XGBoost adds a random decision tree classifier to correct the erroneously classified samples, thus improving the accuracy of data separation. The RF method used on each iteration step of the training process found the best random subset of features for the given classifier to construct the ensemble one [58]. These methods explicitly find most informative features during the training process, so the confidence in the constructed data models is high [59].

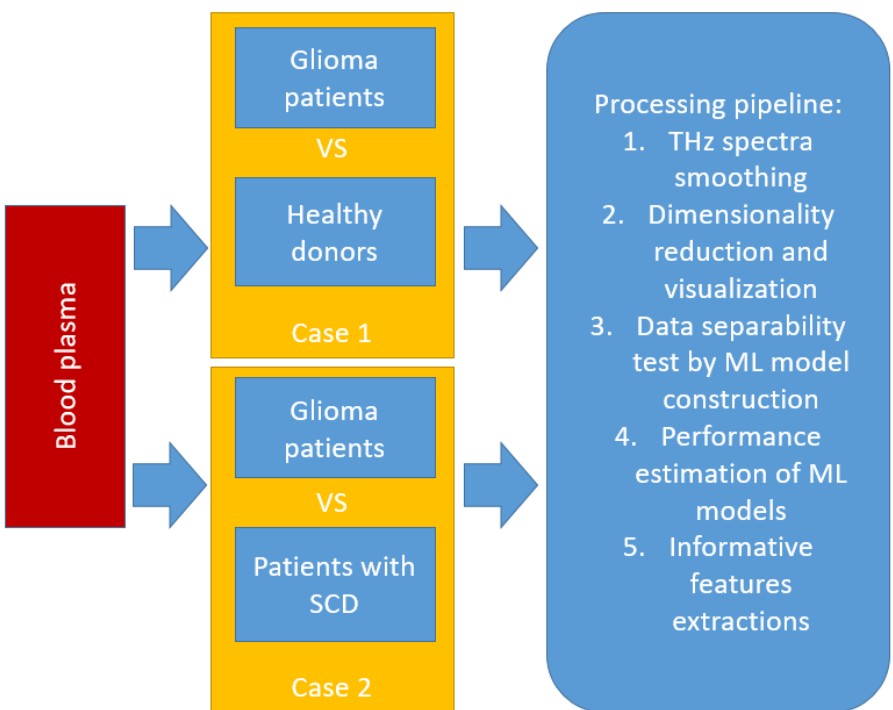

**Figure 2.** Experimental THz data processing pipeline.

ML models' performance was evaluated through a k-fold cross validation technique in terms of ROC-AUC, sensitivity, and specificity metrics. This pipeline has already been applied in our previous works for the analyses of experimental THz and Raman data of mouse blood serum [41,46].

## 3. Results

In Figure 3a we present mean time-resolved THz signals for each group, and in Figure 3b the same graphs but compared to the reference THz signal from an empty cuvette. The time delay and decrease in the signal amplitude when the sample was introduced into the cuvette was associated with the plasma impact in the THz wave transformation. The THz signal for healthy donors was shown to have a smaller amplitude, and was mixed to the right by 0.25 ps relative to the samples of patients with and without glioma, which show similar trends in the overall shape of the curve with a small difference in the THz signal amplitude.

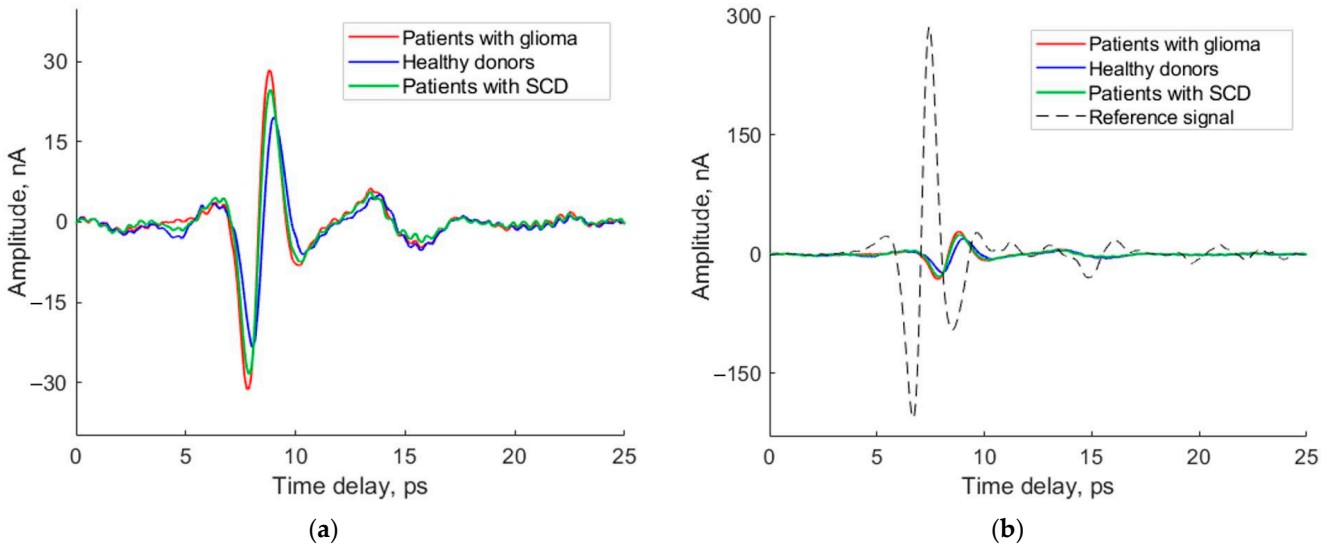

(**a**)　　　　　(**b**)

**Figure 3.** Mean time-resolved THz signals for each group (**a**), and the same graphs but compared to the reference signal (**b**): glioma patients (red line), healthy donors (blue line), patients with SDC (green line), and reference signal (dotted line).

The mean and variance of THz intensity spectra of each group of samples are shown in Figure 4.

At the first stage, the optimal window size ("window_length" parameter) for the Savitzky–Golay filter (scipy.signal.savgol filter package was used) was evaluated using averaged estimates of Sensitivity (SE), Specificity (SP), Accuracy (ACC), Precision (PR) over five-fold cross-validation of SVM, RF, and XGBoost methods. The fitting polynomial order ("polyorder" parameter) was chosen to equal 2, because the third order over-smooths THz spectra, while the first order does not have a significant impact. According to Table 1, the optimal value for window size is 5 because the majority of metrics are higher than for other values.

**Table 1.** Estimation of the "window_length" parameter optimal values for the Savitzky–Golay filter.

| "Window_Length" Value | 5 | 15 | 31 | No Smoothing |
|---|---|---|---|---|
| Average SE of SVM/RF/XGBoost | 0.60/0.62/0.68 | 0.60/0.57/0.58 | 0.70/0.52/0.68 | 0.60/0.57/0.58 |
| Average SP of SVM/RF/XGBoost | 0.47/0.50/0.50 | 0.39/0.47/0.44 | 0.28/0.42/0.42 | 0.60/0.57/0.58 |
| Average ACC of SVM/RF/XGBoost | 0.55/0.57/0.61 | 0.52/0.53/0.53 | 0.54/0.48/0.58 | 0.60/0.57/0.58 |
| Average PR of SVM/RF/XGBoost | 0.65/0.67/0.69 | 0.62/0.64/0.64 | 0.62/0.60/0.66 | 0.60/0.57/0.58 |

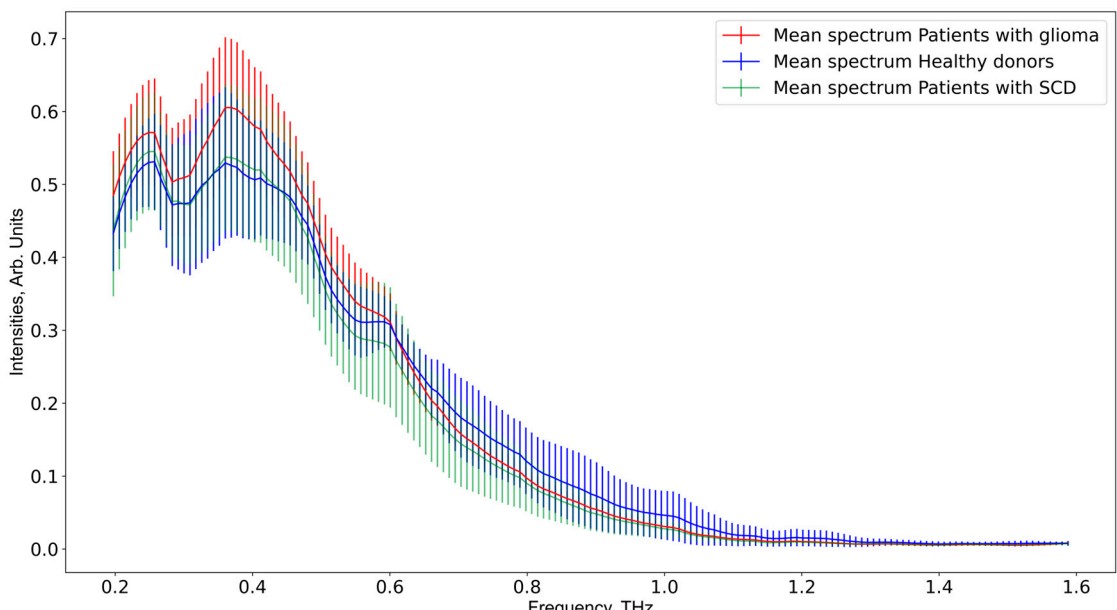

**Figure 4.** Mean THz intensity spectra for blood plasma of glioma patients (red line), healthy donors (blue line), and patients with SCD (green line). All data points have their own bar (vertical line), representing data variance.

At the second stage, data dimensionality reduction was conducted using the PCA and t-SNE methods. In Figure 5, we present the t-SNE visualization. Numbers near markers correspond to the sample ID in each group. Groups are marked by different colors. T-SNE relies on the distance between original spectra, so it does not matter how many groups are under study, contrary to PCA. In Figure 5, we can verify that spectra recorded from one sample are similar. Additionally, tumor and healthy samples form large clusters, but there are outliers.

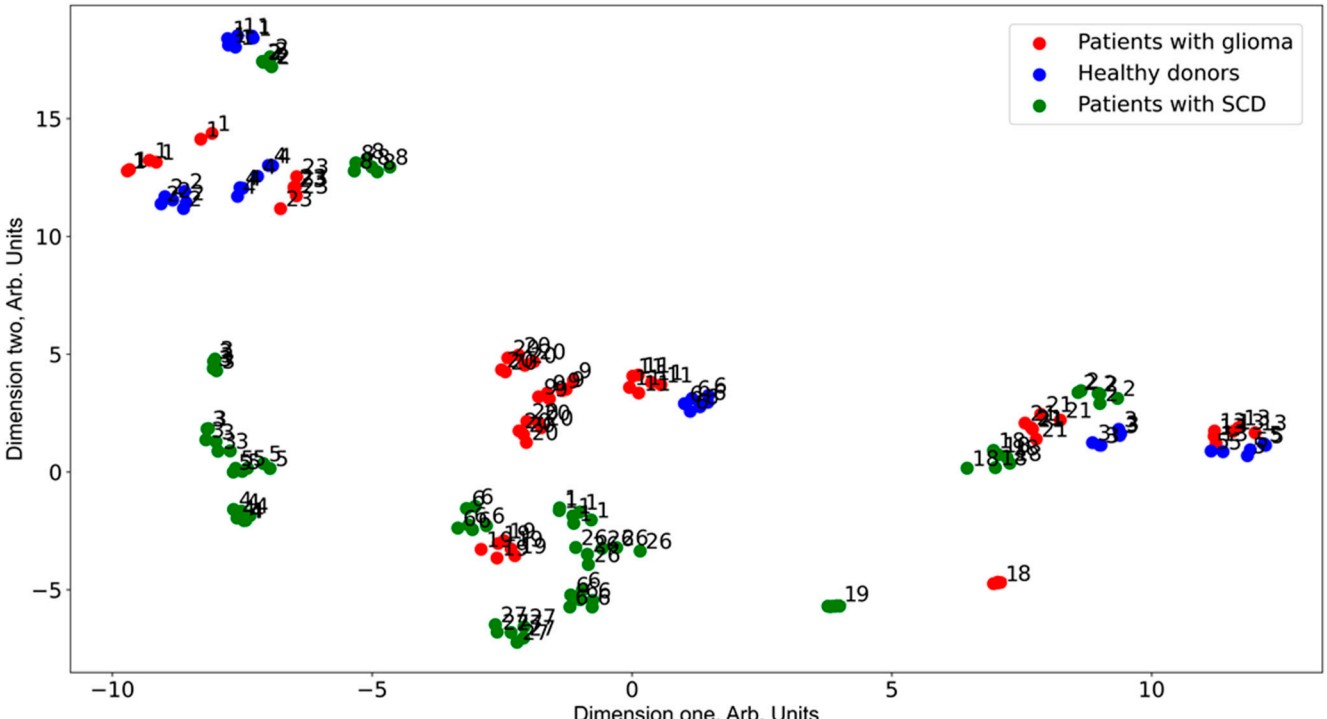

**Figure 5.** T-SNE visualization. Numbers near markers correspond to the sample ID.

The broken stick method for the cumulative explained variance plot is widely used to select an appropriate combination of principal components (PCs) (see Supplementary Materials, Figure S1). A high value of explained variance does not guarantee the best data separation [60]. We were restricted ourselves by using 10 PCs with highest explained variance to achieve an acceptable-to-analyze amount of data (here, the factorial dependence of the number of pairwise combinations on the number of PCs), and to take into account components with a large explained variance, since they often contain important information. Additionally, we tested PCA with and without data standardization and found that, without standardization, it performs better. A possible reason for this is that the THz spectra of studied groups differ in amplitude.

Graphs of the cumulative explained variance to find the maximum allowed number of PCs for the blood plasma of glioma patients and donors, and the blood plasma of glioma patients and patients with SCD, are given in the Supplementary Materials (Figure S1). A steeper curve indirectly indicates that the first PCs contain more valuable data. The smoother ones indicate an even contribution of the PCs and the need to analyze more combinations of them.

PCA score plots for different pairs of studied groups are presented in Figure 6. There was no perfect separation of groups, and the most difficult data for linear separation corresponded to Case 1 (glioma patients versus healthy donors). Additional figures, illustrating the impact of data standardization and the choice of PC combinations, are presented in the Supplementary Materials (Figures S2–S4). This means that two dimensions, provided by PCA and t-SNE for visual analysis, are not sufficient to differentiate groups under study.

At the third stage, the separability of the data was analyzed using a predictive model based on the linear kernel SVM, XGBoost, and RF. The classifier quality was estimated using a Receiver Operating Characteristics (ROC) curve and area under the curve (AUC) analysis. The ROC curve plotted the sensitivity vs. (1-specificity), and the AUC displayed the area under the ROC. Details about ROC-AUC analysis and corresponding ROC-AUC curves are presented in the Supplementary Materials (Figure S5). The results are summarized in Table 2. Here, the AUC value was averaged over all training and testing set implementations according to the 10-fold cross-validation and presented in terms of mean and standard deviation values.

**Table 2.** Averaged AUC for Cases 1 and 2 and linear kernel SVM, RF, and XGBoost classifiers.

| Case Number | Averaged AUC, Linear Kernel SVM | Averaged AUC, RF | Averaged AUC, XGBoost |
|---|---|---|---|
| Case 1 | $0.72 \pm 0.3$ | $0.82 \pm 0.25$ | $0.82 \pm 0.21$ |
| Case 2 | $0.70 \pm 0.3$ | $0.84 \pm 0.20$ | $0.81 \pm 0.25$ |

As can be seen from Table 2, the AUC mean value was higher for the XGBoost and RF data models, and simultaneously their AUC variance was lower compared to that of linear kernel SVM. This means that XGBoost and RF show the best separation of the group of patients with glioma from healthy donors and patients with SCD. At the final stage, informative frequencies were identified. The results are shown in Figure 7.

According to Figure 7, the informative frequencies differ for various groups, although there is also an intersection, often with different contributions for cases 1 and 2. Tree-based ensemble methods were shown to work more accurately than the SVM method. However, the informative frequencies may not coincide, which indicates the instability of the methods. This is owing to the heterogeneity of the data of patients with gliomas, and we will obtain more robust ML models with an increasing amount of samples.

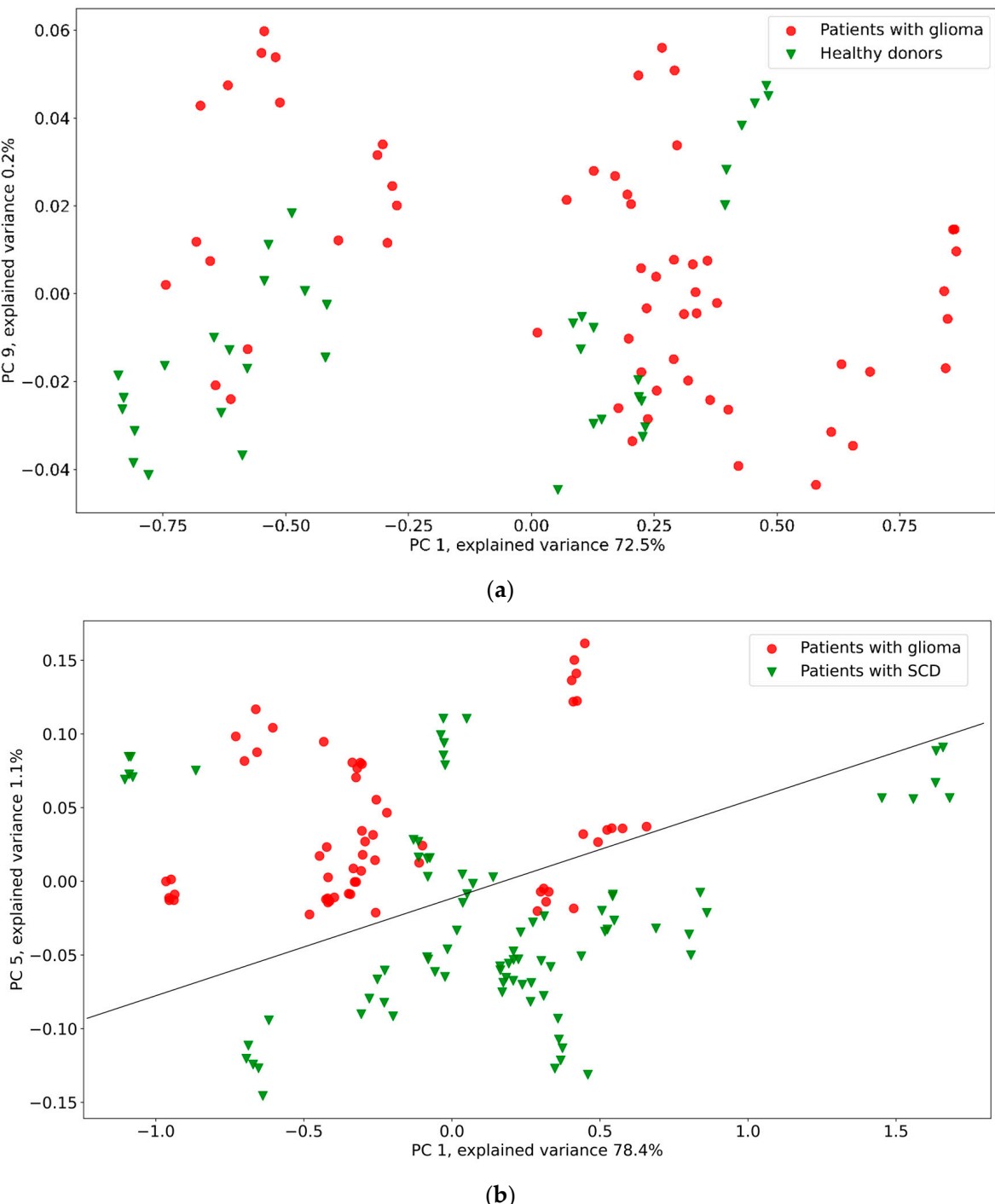

(**a**)

(**b**)

**Figure 6.** PCA score plots for Case 1 (glioma patients versus healthy donors, (**a**)) and Case 2 (glioma patients versus patients with SCD, (**b**)). Linear separation border between classes is shown for Case 2.

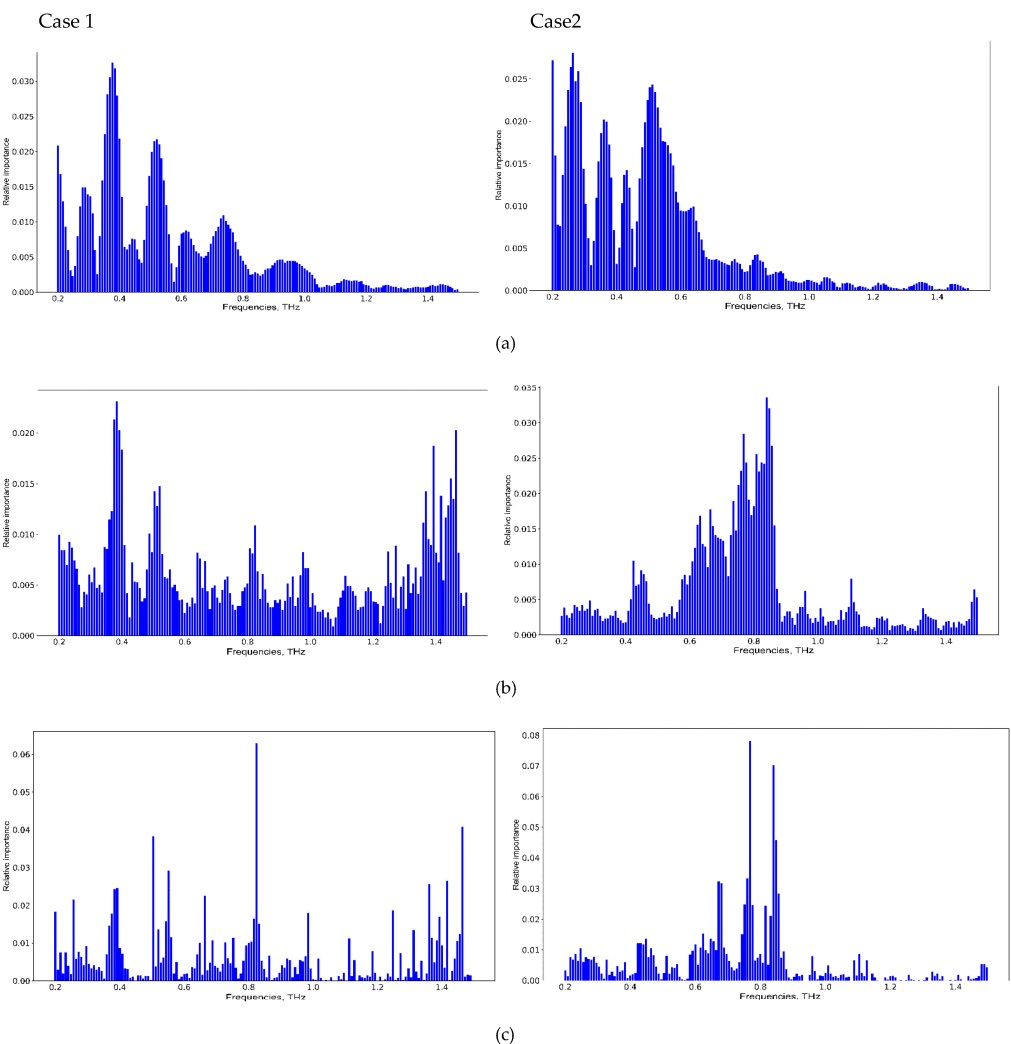

**Figure 7.** Informative frequencies for Case 1 and Case 2 with different classifiers: (**a**)—linear kernel SVM; (**b**)—RF; (**c**)—XGBoost.

Among the patients with glioma, there were five pairs of patients whose blood plasma samples were taken before and a week after the surgical intervention to remove the tumor. ML methods were applied to establish differences in the THz spectra before and after the surgery. It can be assumed that some specific features of an individual patient were leveled and differences in the spectra may be associated with the presence or absence of the tumor. Figure 8 shows the mean and variance of the THz spectra of glioma patients before the surgery and 7 days after the surgery. The graph of the cumulative explained variance to determine the maximum allowable number of PCs is presented in the Supplementary Materials (Figure S6). It shows that it is not only the first PCs that should be checked for the better separation of the studied groups. The mean AUC for the blood plasma of glioma patients before and after the surgery were as follows: for linear kernel SVM: 0.50, for RF method: 0.92, and for XGBoost: 0.86. Corresponding images for ROC-AUC are given in the Supplementary Materials (Figure S7).

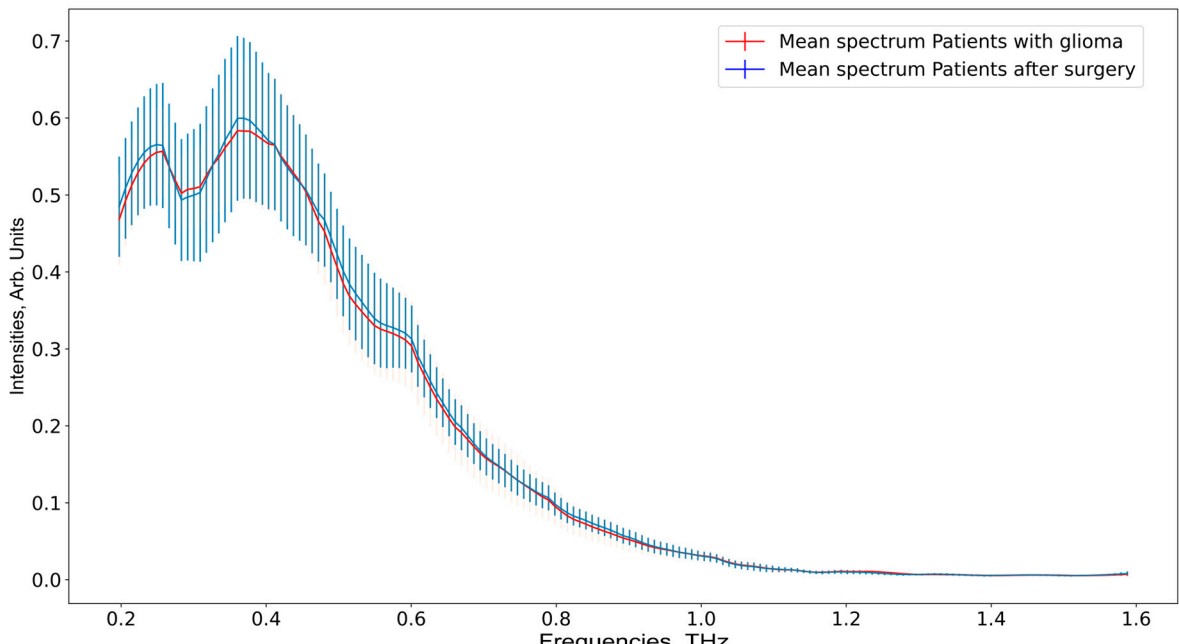

**Figure 8.** The mean THz spectra of blood plasma of glioma patients before (blue line) and 7 days after (red line) the tumor removal surgery. All data points have their own bar (vertical line), representing data variance.

According to Figure S7, RF and XGBoost methods made it possible to distinguish the blood plasma of glioma patients before and after surgery with good discrimination (AUC = 0.86–0.92). The informative frequencies allowing the separation of these groups are shown in Figure 9.

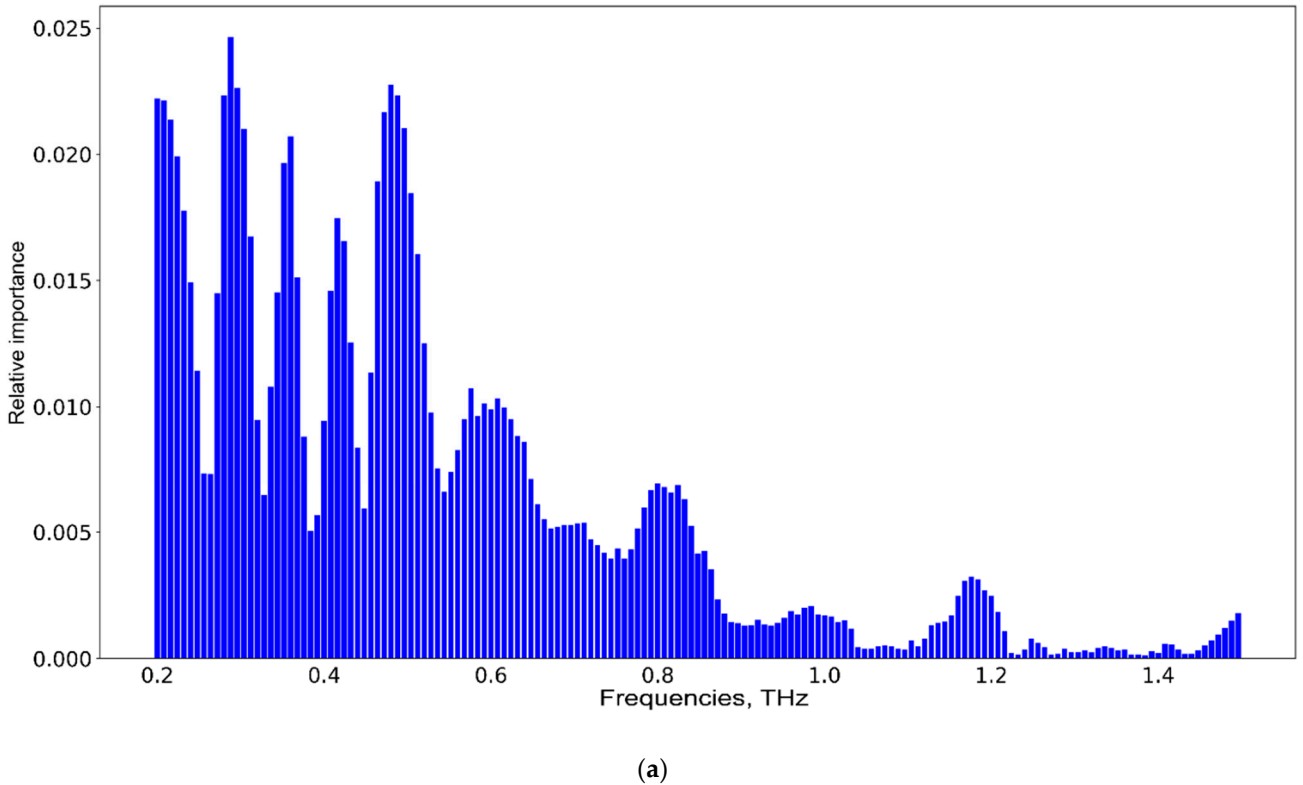

(**a**)

**Figure 9.** *Cont.*

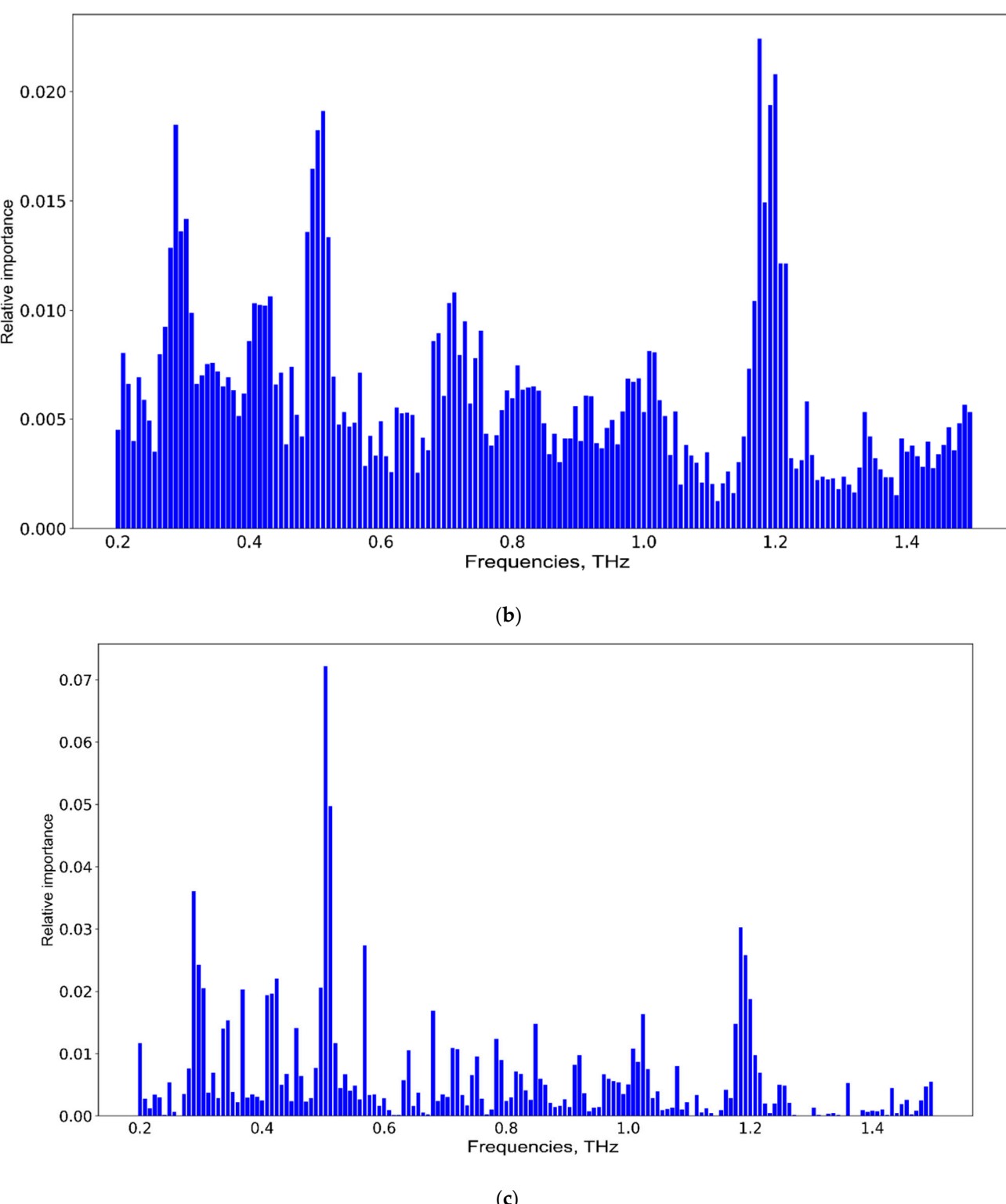

(**b**)

(**c**)

**Figure 9.** Informative frequencies allowing the distinction between glioma patients before and after the surgery by the following classifiers: (**a**)—SVM; (**b**)—RF; (**c**)—XGBoost.

According to Figure 9, the most informative frequencies are 0.56 and 1.2 THz. Though they did not coincide with the frequencies with the highest variance, both RF and XGBoost methods considered them to be the most informative ones. The possible reason is that these methods tested separability by each frequency, disregarding the absolute value of the data variance.

## 4. Discussion

The blood of cancer patients contains tumor molecular biomarkers [21–23], but their identification is not a trivial task [61]. THz spectroscopy is sensitive to the presence of biomolecules in a sample [30]. For example, a correlation of THz absorption with concentrations of serum glucose [36,38] and protein [30,40] has been demonstrated. However, molecules' absorption bands overlapping in the THz spectral range make it difficult to discover specific metabolites in blood samples (both serum and plasma) by conventional statistical analysis due to correlations between frequencies and the high dimensionality of the data [62]. Machine learning methods allow these problems to be overcome, build predictive data models, and discover informative sub-bands in THz spectral range [36,38,39,41,63].

In this work, we applied PCA and t-SNE dimensionality reduction methods to the THz spectral data of human blood plasma. According to the analysis conducted using t-SNE, the measured THz spectra from a sample are close to each other in the feature space, which indicates the acceptable homogeneity of the THz data for an individual blood plasma sample. At the stage of predictive data model construction, both XGBoost and RF methods were shown to work more accurately than SVM: they provided a better separation of the glioma patients' group from healthy controls and patients with SCD.

Our methods can be compared to the ones proposed in [42]. In this paper, rat blood serum at different degrees of experimental blast-induced traumatic brain injury was studied by THz-TDS and ML methods including PCA and k-Nearest neighbor (kNN) and SVM algorithms. In our previous study, we also studied small animal models and the classification accuracy of the SVM classifier measured by ROC-AUC analysis was good [41]. Instead of using the *p*-value, which has recently been criticized, we prefer ML methods such as Isolation Forest, PCA, and t-SNE for visual analysis. Another option that is missed is data standardization, usually recommended prior to PCA. We studied this topic, and it has demonstrated that THz data standardization can decrease the performance of the classifier. Performance metrics such as accuracy, sensitivity, and specificity are usually presented as mean $\pm$ standard deviation (STD), because training models on small dataset can lead to drastic variance due to outliers in groups. Therefore, we prefer mean ROC-AUC with an STD interval.

The major challenge is the non-invasive control of the effectiveness of the tumor removal surgery [64]. According to the presented results, RF and XGBoost predictive data models made it possible to distinguish the blood plasma of glioma patients before and after surgery with good discrimination: AUC was 0.92 and 0.86, respectively. The SVM did not show a satisfactory result (AUC = 0.5). The most informative frequencies were 0.56 and 1.2 THz. These frequencies coincided with informative frequencies obtained in an experimental mouse model of human glioblastoma [41]. However, an unresolved problem was finding correspondence between metabolites in blood plasma and informative THz frequencies. A positive moment was that we discovered an informative THz range for glioma diagnostics, and it was valuable information for making cheaper devices.

## 5. Conclusions

In this work, we studied the blood plasma of patients with glioma using THz-TDS and ML. We carried out a comparative analysis of the THz spectral data of this group with a group of patients with SCD and healthy donors. The XGBoost and RF predictive data models showed the best differentiation of the glioma patients' group from the other groups (AUC = 0.81–0.84).

It was shown that THz spectroscopy combined with ML makes it possible to separate with high accuracy the blood plasma of glioma patients before and after tumor removal surgery. The most informative THz frequencies were determined. They coincided, for these, in the experimental mouse model of human glioblastoma found by us earlier [41]. This confirms the potential of THz spectroscopy and ML in the control of glioma treatment.

The main limitation of this study is the number of processed samples. We believe that more robust ML models can be constructed by increasing the dataset volume.

**Supplementary Materials:** The following supporting information can be downloaded at: https://www.mdpi.com/article/10.3390/app13095434/s1, Figure S1: Graph of the cumulative explained variance to find the maximum allowed number of PCs for (a) blood plasma of glioma patients and healthy donors (Case 1); (b) blood plasma of glioma patients and patients with skull craniectomy defects (Case 2); Figure S2: PCA score plot with data standardization (a) and without (b) for glioma patients and healthy donors, for PC1 and PC2; Figure S3: PCA score plot with data standardization (a) and without (b) for glioma patients and healthy donors, for PC1 and PC9; Figure S4: PCA score plot with data standardization (a) and without (b) for glioma patients and patients without glioma (with skull craniectomy defects), for PC1 and PC5; Figure S5: ROC AUC analysis: (a)—linear kernel SVM Case 1; (b) linear kernel SVM Case 2; (c)—RF Case 1; (d)—RF Case 2; (e)—XGBoost Case 1; (f)—XGBoost Case 2; Figure S6: A graph of cumulative explained variance for determining the acceptable number of principal components; Figure S7: Graph of ROC-AUC analysis of different ML methods for separation of blood plasma of glioma patients before and after surgery: (a)—linear kernel SVM; (b)—RF; (c)—XGBoost.

**Author Contributions:** Conceptualization, O.C., Y.P., Y.K. and A.S.; methodology, Y.K., Y.P., D.V., M.K., N.N. and A.K.; software, D.V. and A.P.; validation, D.V., A.K., A.P., M.K., N.N., E.S., V.G. and V.S.; formal analysis, D.V., A.K., E.S., M.K., V.G., V.S. and N.N.; investigation, A.K., E.S., V.G., A.P. and N.N.; writing—original draft preparation, O.C., Y.P., D.V., A.K., E.S. and V.S.; writing—review and editing, O.C., Y.P., Y.K., D.V., A.K., A.P., E.S., V.G., V.S., N.N., M.K. and A.S.; visualization, D.V., A.K. and A.P.; supervision, A.S.; project administration, O.C. and Y.K.; funding acquisition, N.N., Y.K. and A.S. All authors have read and agreed to the published version of the manuscript.

**Funding:** This research was funded by the Ministry of Science and Higher Education of the Russian Federation: project No. 121032400052-6. This work was performed partly within the State Assignment of FSRC "Crystallography and Photonics" RAS. This work has been supported by the Interdisciplinary Scientific and Educational School of Moscow University «Photonic and Quantum Technologies. Digital Medicine» for part of the results analysis. The research was carried out with the support of a grant under the Decree of the Government of the Russian Federation No. 220 of 9 April 2010 (Agreement No. 075-15-2021-615 of 4 June 2021). The work of D.V., A.K. and Y.K. was supported by the Ministry of Science and Higher Education of the Russian Federation (budget funds for the V.E. Zuev Institute of Atmospheric Optics of the Siberian Branch of the Russian Academy of Sciences).

**Institutional Review Board Statement:** The study was conducted according to the guidelines of the Declaration of Helsinki, and approved by The Ethics Committee of the Novosibirsk Research Institute of Traumatology and Orthopedics n.a. Ya.L. Tsivyan (Protocol #004/22-1, 17 January 2022).

**Informed Consent Statement:** Informed consent was obtained from all subjects involved in the study.

**Data Availability Statement:** The data presented in this study are available on request from the corresponding author.

**Conflicts of Interest:** The authors declare no conflict of interest. The funders had no role in the design of the study; in the collection, analysis, or interpretation of data; in the writing of the manuscript; or in the decision to publish the results.

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
