# Peer review of "Terahertz Time-Domain Spectroscopy of Glioma Patient Blood Plasma: Diagnosis and Treatment"

_applsci, doi:10.3390/app13095434_

Round 1

Reviewer 1 Report

In the current study, the authors aimed to integrate a machine-learning pipeline to analyze the output of terahertz time-domain spectroscopy for the diagnosis of glioma. I have some comments about this manuscript:

1- The weakest section in the manuscript is the discussion where the authors repeated the findings in their results and repeatedly returned us back to the results figures. I can not find a comparative analysis between the current study and previous correlated studies therefore the quality of the discussion needs improvement.

2- What are the limitations and the error rate of the current study approach? 

3- Minor comments: - An introductory sentence should be added at the beginning of the abstract.

- The first sentence in the discussion is poorly written. Kindly rephrase.

- Some figures' quality (especially figures 7, 8, and 11) should be improved.

Moderate editing of English language is required

Reviewer 2 Report

The manuscript describes the study of the blood plasma of patients with glioma and without glioma, as well as healthy donors using the terahertz time-domain spectroscopy (THz-TDS) combined with machine learning pipeline (MLP) for data analysis, in an attempt to demonstrate the capability of THz-TDS and machine learning for the diagnosis of glioma and treatment monitoring.

The topic of applying THz-TDS and machine learning as a glioma diagnosis and treatment monitoring tool is novel and of clinical significance, and the application of machine learning may potentially increase the efficiency of the diagnosis in terms of accuracy and time. However, what advancement and novelty this MS provides beyond the author’s previous works published in references 2,3,6-8,11 is not clear.

The Introduction does not provide sufficient background of the THZ-TDS technique for uninitiated readers to follow. And the brief introduction of references 6,7 leaves readers puzzled. It begs the question: are the authors the first and only group that applies THz-TDS in cancer diagnosis? How about in the wider biomedical field?

There is a missing link between the conclusion and the applicability of THz-TDS and machine learning for the diagnosis of glioma and treatment monitoring. Please explicitly explain this, how will the results be applied to diagnosis? Are the proposed methods still under investigation and validation and how far are they from being applied clinically?

Thus a major revision is recommended before further consideration.

Some suggestions:

1. To remedy the lack of background of the THZ-TDS technique, the authors may refer to doi 10.1039/d1ma01002f, 10.1038/ncomms9631 where these have been introduced and explained in the context of biomaterials for medical applications.

2. What are the model and serial No. of the THZ-TDS? Please provide these for reproducibility. What are the spectral resolution, acquisition rate and dynamic range?

3. What is the objective of using the special cuvettes? What are their absorbance compared to normal cuvettes? Did you do parallel measurements without cuvettes and empty cuvette for comparison, also for subcontracting the cuvette transmission?

4. Are the data points in Fig. 3 mean and error bars? Is each line the spectrum of one selected sample of each group? The text and figure caption do not reflect this though. This also applies to Fig. 9. How about the time-resolved spectra? What do they look like?

5. What does “most difficult data” mean?

6. Fig. 7 and 8 are of low resolution.

7. L216, how will you “get more stable results”?

8. “These frequencies coincide with informative frequencies obtained in an experimental mouse model of human glioblastoma” should be moved up to Discussion of the result.

9. It appears that the Discussion section is missing, at least to put your results in the wider cancer diagnosis context.

10. Some terminology is not consistent, e.g. “healthy volunteers” and “healthy donors” are both used. What are “patients with injuries”?

11. Fig. 4, are all samples included? E.g., for healthy donors, n=11, how come in Fig. the IDs are only 1-6?

12. Please explain better how the conclusion “As can be seen from Figure 7, the XGBoost method and RF show the best separation of the group of patients with glioma from healthy donors and patients without glioma” was drawn.  which panel, which feature was the conclusion based on?

13. Although the machine learning results lead to “the most informative frequencies are 0.56 and 1.2 THz,” seen from Fig. 9, at 1.2 THz blue (although it spears green in plot) and red spectra are not significantly different. On the contrary, seen from Fig. 3, 9 around 0.4 THz seems to the range where the most differences are. Why is this? Also what possible biological features are possibly manifested at 0.56 and 1.2 THz frequency, do we know? What are the fundamental differences between blood plasma of patients with glioma and without glioma? Some biological relevance may be useful here.

Round 2

Reviewer 1 Report

The authors have addressed my comments and I have no other concerns.

Moderate editing of English language is required.

Author Response

We would like to thank you for your work with our submission and for your thoughtful and relevant remarks. We have revised our manuscript according to your comments. 

Reviewer 2 Report

The authors have addressed some issues, some satisfactorily, some unsatisfactorily.

Please explain which plots show "We can differentiate the blood of patients with glioma from other patients and healthy people". Fig. 7 certainly does not show this. How did you impress the practicing neurosurgeons? Please provide the same evidence-based arguments to impress/convience the reviewers and readers.

Fig. 1 is not necessary since this is not a result nor original design and the set up is very simple.

"Vertical lines correspond to the variance of the data. Each data point has own bar, representing data variance." appears repetitive.

Fig. 8 is blurred, Fig. 12 is overly big. Also it is not clear what these 2 figs present, nor how they lead to the overall conclusions. The same applies to Fig.s 6 and 11. 

The explanation of the ML is poor and hard to follow.

Some corrections are needed:

L407, take out , in front of that ", that"

L439, with is normally used in combination with coincide

Please refine "This THz-TDS is equipped a micro-strip photoconductive antenna fabricated on low-temperature grown GaAs substrate ensures dynamic range >90 dB at 0.4 THz."

equip...with, there are 2 verb.s
